# Emergence and control of photonic band structure in stacked OLED microcavities

David Allemeier[1], Benjamin Isenhart[2], Ekraj Dahal [1], Yuki Tsuda [3], Tsukasa Yoshida [4] & Matthew S. White [1,2]✉

We demonstrate an electrically-driven metal-dielectric photonic crystal emitter by fabricating a series of organic light emitting diode microcavities in a vertical stack. The states of the individual microcavities are shown to split into bands of hybridized photonic energy states through interaction with adjacent cavities. The propagating photonic modes within the crystal depend sensitively on the unit cell geometry and the optical properties of the component materials. By systematically varying the metallic layer thicknesses, we show control over the density of states and band center. The emergence of a tunable photonic band gap due to an asymmetry-introduced Peierls distortion is demonstrated and correlated to the unit cell configuration. This work develops a class of one dimensional, planar, photonic crystal emitter architectures enabling either narrow linewidth, multi-mode, or broadband emission with a high degree of tunability.

[1] Materials Science Program, University of Vermont, Burlington, VT 05405, USA. [2] Department of Physics, University of Vermont, Burlington, VT 05405, USA. [3] Graduate School of Science and Engineering, Yamagata University, Yonezawa, Yamagata 992-8510, Japan. [4] Graduate School of Organic Materials Science, Yamagata University, Yonezawa, Yamagata 992-8510, Japan. ✉email: mwhite25@uvm.edu

Organic light emitting diodes (OLEDs) constitute a highly adaptable lighting technology with widespread application in consumer electronics. OLED technologies have proven themselves well-suited to hybrid materials systems which integrate organic semiconductor materials, dielectrics and metals to enable scientifically and technologically interesting device architectures and functionality[1–5]. One such hybrid system is the Fabry-Pérot microcavity OLED, which consists of an active OLED stack bounded by semitransparent metal mirrors[6–12]. These devices allow for precise tuning of narrow linewidth emission from OLEDs, largely independent of the chosen emitter material[6,13,14]. In this work, we form a vertical stack of such cavities to produce an alternating metal-dielectric photonic crystal (MDC) structure whose unit cells consist of one or more electrically-driven OLED microcavities. The resulting photonic band structure is directly revealed through the electroluminescence spectrum. We show the emergence of photonic energy bands as the number of unit cells is increased and demonstrate control of the density of states and intra-band bandgap by tuning physical device parameters.

In an optical microcavity, or Fabry-Pérot etalon, the presence of two parallel metallic mirrors requires the internal electric field to have nodes at the boundaries. For real metals, this condition is relaxed and the electric field decays into the mirrors with a characteristic penetration depth $\phi$. The resulting resonance condition is thus expressed[15]:

$$\lambda_j = \frac{2}{\left(j - 2\left|\frac{\phi_\uparrow + \phi_\downarrow}{4\pi}\right|\right)} \sum_i n_i t_i \qquad j = 1, 2, 3 \dots \tag{1}$$

where $\lambda_j$ is the resonant wavelength of index $j$, $\phi_\uparrow$ and $\phi_\downarrow$ are the phase shift at the top and bottom mirrors, and $n_i$ is the index of refraction of each layer of thickness $t_i$ between the mirrors. This forms a discrete series of resonant wavelengths with the longest (lowest energy) mode being roughly twice the optical pathlength between the mirrors. Emission from within such a cavity inherits these structural photonic modes, narrowing the emission linewidth and enhancing the spontaneous emission through the Purcell effect[16,17].

In this work, we demonstrate that when multiple microcavities are placed in contact, as depicted in Fig. 1, the penetration of the resonant electric field through the metallic mirrors allows interaction between adjacent cavities. This interaction results in the single-cavity photonic states splitting into a band in a process exactly analogous to the formation of electronic energy levels in a crystal or hybridized states in a molecule. Consideration of the two limiting cases in Fig. 1a elucidates the role of the mirrors in the formation of the photonic bands. When the thickness of the mirrors greatly exceeds the penetration depth, no communication is possible between the cavities and they behave as in isolation (Fig. 1a on the left). If the internal mirrors are infinitely thin, however, there is simply a single extended microcavity between the top-most and bottom-most mirrors. At intermediate thicknesses, on the order of the penetration depth, the internal electric field is able to partially penetrate the barriers between adjacent cavities. The internal mirrors act to perturb the extended cavity states, as shown in Fig. 1b, and the resulting interaction results in the emergence of photonic energy bands centered at the isolated cavity states. The distribution of states within the bands depends sensitively on the unit cell geometry and materials. As we show, this allows a high degree of control over the position, separation, and spectral profile of the photonic states of the crystal.

## Results

**Emergence of photonic bands.** The optical microcavity has deep connections to many sub-fields of physics and represents a special case of generalized cavity-wave phenomena. The properties of such cavities share many of the same features as the quantum finite square well, including the emergence of discrete energy states and particle tunneling[18]. The periodic extension of the square well concept forms the basis of the Kronig-Penney model of solid-state physics and provides solutions for the electronic states within a metallic crystalline lattice. In covalently bonded systems, the electronic states are modeled by the closely related linear combination of atomic orbital (LCAO), molecular-orbital (MO) and Hartree-Fock theories. Together, these theories explain the emergence of electronic energy bands from discrete atomic and molecular energy levels. The central result is that when N identical systems are brought into contact, each initial energy state of the isolated systems will split into N hybridized energy states. The separation between the hybridized states, and the existence of band-gaps, depends on the magnitude and shape of the potential and hence the physical parameters of the system.

The MDC system presented here differs from these models in two critical ways. First, the confined particles are photons and hence obey bosonic rather than fermionic statistics. Second, the periodic structure is absorbing, which manifests as a complex, rather than purely real, potential. The eigenvalues of the unit cell matrix that describes the photonic crystal therefore take on Floquet, rather than purely Bloch form, wherein translation of the solutions across unit cells requires an attenuation coefficient[19,20]. This is distinct from a lossless dielectric photonic crystal (DPC), wherein the propagation constant is unity[21]. The attenuation experienced by the propagating wave is highly dependent on the effective reflectivity of the structure, which in turn is modulated by the Fabry-Pérot microcavity geometry[22]. This has led to the study of similar MDC structures in the context of optical filtering to enhance the transmission and reflectivity of metallic films[19,22].

For our study, a standard OLED stack served as the dielectric layer within the MDC to enable direct observation of the photonic band structure within the crystal. The six-cavity structure depicted in Fig. 1c was selected as the baseline for experimentation, characterized by 30 nm mirrors, a 100 nm base mirror, and 115 nm OLED stacks. The OLED stack within each cavity was kept constant for all experiments and consisted of 35 nm of NBPhen as the electron transport layer (ETL), 10 nm of Alq$_3$ as the emitter layer (EML) and 70 nm of NPB as the hole transport layer (HTL), with 1 nm of MoO$_x$ and LiF as the hole and electron injecting layers, respectively. From previous work, this was expected to produce an emission peak at 560 nm (2.21 eV) for a single cavity and thereby define the band center[6]. This ensures maximum overlap of the states with the electroluminescence spectrum of Alq$_3$ as we increase the bandwidth. The mirrors are expected to show surface plasmon resonance at ~390 nm (Ag) and ~ 800 nm (Al), which bracket the electroluminescence of Alq$_3$, but this phenomenon must be considered when shifting to the blue/violet or the IR. Other material combinations may allow for surface plasmon polariton modes to further shape the emission spectra. The cavities were electrically pumped in parallel, with shared anodes composed of a 15:1 Ag-Al alloy[23] and cathodes of pure Al. The parallel-driven electrical architecture allows lower driving voltages and higher operating stability but also requires flipping the OLED stack within each cavity. In combination with the alternating mirror composition, this leads to a natural doubling of the unit cell length. The unit cells therefore consist of two microcavities of opposite orientation as shown in Fig. 1d, a feature which is apparent in the contrast between the metal mirrors when viewed by cross-sectional SEM in Fig. 1e.

The first component of our study demonstrates the emergence of the photonic band structure from the single cavity state. Six devices were fabricated with $N = 1$ to $N = 6$ cavities and their

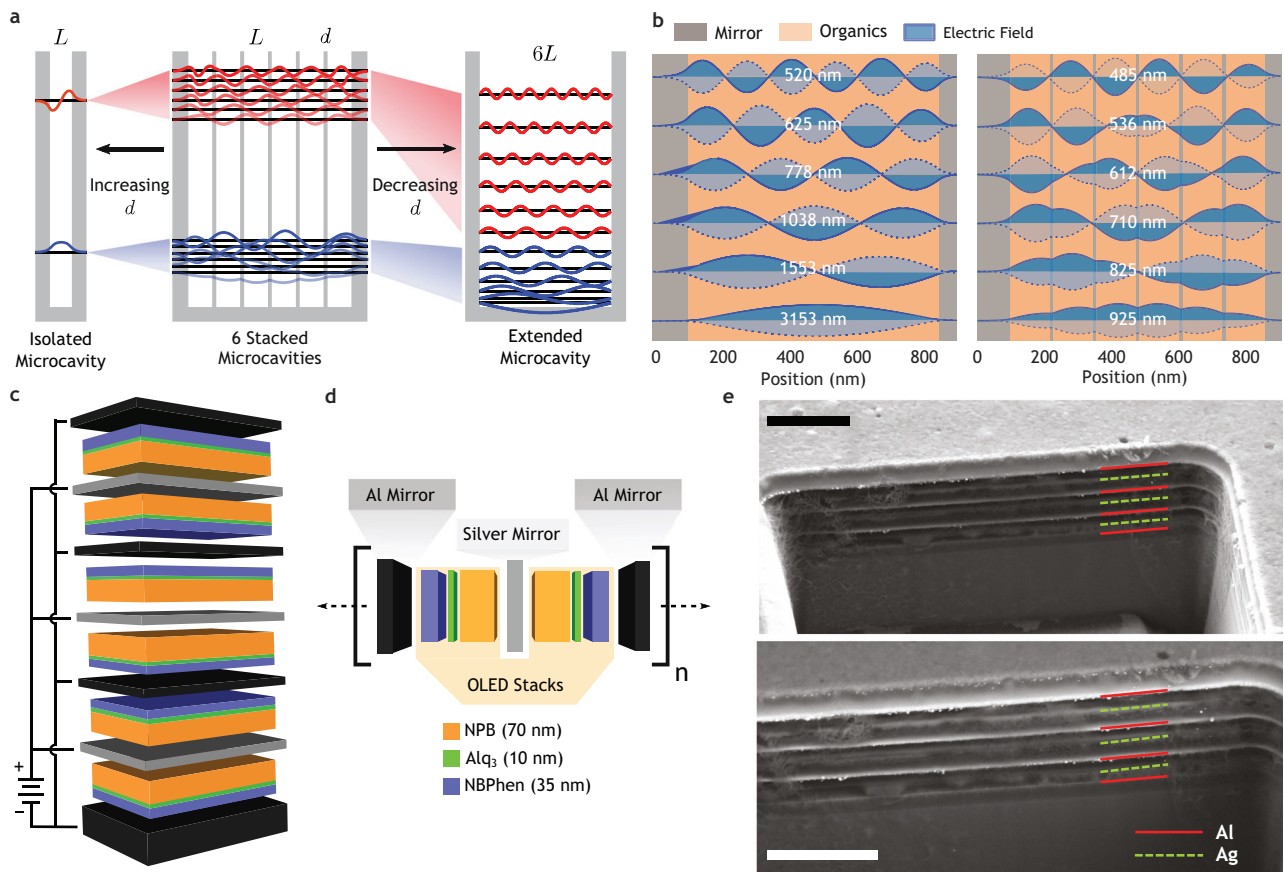

**Fig. 1 Stacked microcavity OLEDs. a** Schematic showing the formation of energy bands through the interaction of the isolated single-cavity and extended cavity modes. Increasing the mirror thickness reduces the bandwidth, compressing the states to those of the single-cavity. Reducing the mirror thickness increases the bandwidth, and the states approach the extended cavity modes. **b** Transfer matrix simulations of the internal electric field profiles at normal incidence of the first photonic band of an ideal extended cavity of length $6L = 720$ nm (left) and an ideal six-layer stacked microcavity device with $L = 120$ nm and $d = 10$ nm (right). Perturbation of the states by the internal mirrors increases their energy. **c** Device schematic of a six-layer stacked microcavity device with alternating Al and Ag mirrors of uniform 30 nm thickness. The microcavities are electrically driven in parallel with shared anodes and cathodes. **d** Schematic of a single unit cell of the photonic crystal consisting of two identical OLED devices with opposite orientation. Mirror thicknesses were varied between 10 and 30 nm. NPB (N,N'-Bis(naphthalen-1-yl)-N,N'-bis(phenyl)-benzidine), Alq$_3$ (Tris-(8-hydroxyquinoline)aluminum, NBPhen (2,9-Bis(naphthalen-2-yl)-4,7-diphenyl-1,10-phenanthroline). **e** Cross-sectional SEM image of an $N = 6$ device taken at 45° tilt prepared by focused ion beam, using secondary electron (top) and backscatter (bottom) detectors. Scale bars 1 μm.

electroluminescence spectrum was measured. Transfer matrix simulations were conducted to model the expected emission spectrum of the devices. Discussion of the simulation may be found in the Supplementary Information (SI), and further details of the theory are provided in the accompanying SI of Dahal et al.[6] Unless otherwise noted, the simulated results represent the raw expected peak positions without accounting for thickness variations or surface roughness. The normal incidence emission spectra of the stacked microcavity devices from $N = 1$ to $N = 6$ are shown in Fig. 2a. The single emission peak for the $N = 1$ device was observed at 554 nm, matching our target of 560 nm. This roughly defined the center of the photonic band for subsequent experiments. As discussed further in the SI, however, the band center is also a function of the mirror thickness due to the effect on the overall optical pathlength of the structure. Small variations in the deposited film thicknesses resulted in the true band center fluctuating for subsequent devices and affected the distribution of states within the bands. This effect is the primary source of error between the simulated and observed results. The relatively broadband emission from the devices inhibited resolution of some modes, particularly the fundamental (lowest energy) mode. Further refinements to the microcavity design and

fabrication techniques can improve the accuracy and finesse of the states and the device efficiency[6,7].

The emergence of the photonic band was observed in Fig. 2 as the number of cavities $N$ was increased. At $N = 2$, the single-cavity peak at 554 nm split into two peaks, one at higher and one at lower energy. The central peak observed in the $N = 2$ device was not predicted by simulations and coincides with the single-cavity emission. This represents edge-effect leakage from the single cavities due to slightly misaligned shadow masks. The presence of three peaks was confirmed by multi-peak fitting; the outer two of which correspond to the crystalline states. Edge-effect leakage was observed in several samples but is most apparent for devices with even numbers of cavities. Odd-numbered devices have overlapping states with the single cavity mode and therefore any leakage simply magnifies the signal of this central state, as seen in Fig. 2b.

The crystalline nature of the stacked OLED structure becomes apparent when the number of unit cells is at least two. For the $N = 1 \rightarrow 3$ devices, the crystal is poorly defined and is more accurately described by a single unit cell composed of $1 \rightarrow 3$ microcavities. This results in the even distribution of states observed in Fig. 2b. The addition of a fourth cavity completes the

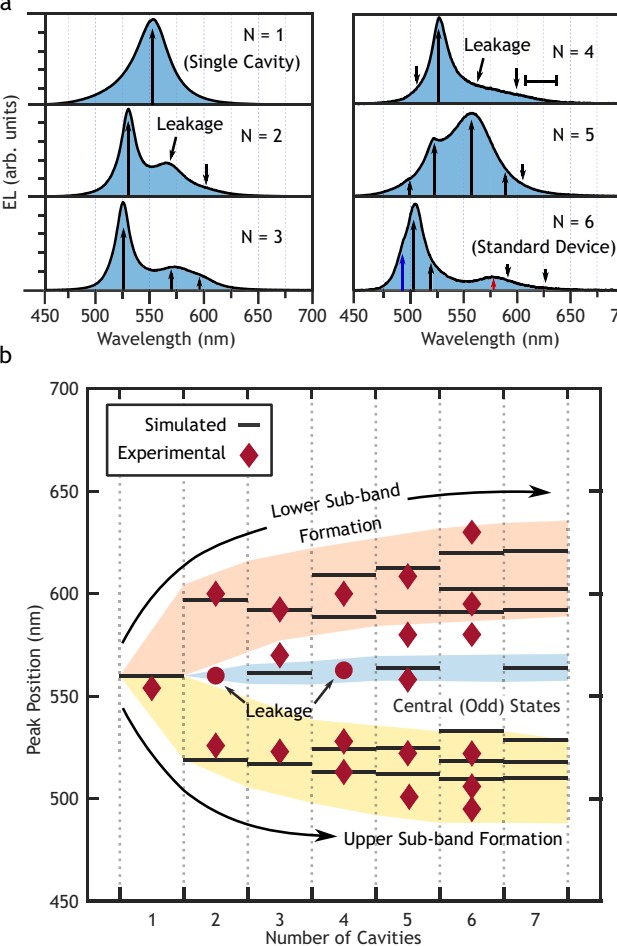

**Fig. 2 Emergence of photonic bands. a** Electroluminescence spectra at normal emission for $N = 1 \rightarrow 6$ stacked microcavity OLEDs. The highest energy peak at 495 nm had a linewidth of 10.5 nm (blue arrow) while the highest energy peak in the lower sub-band at 578 nm had a linewidth of 47 nm (red arrow). **b** Computationally simulated and experimentally resolved wavelength of the center of each state in the photonic bands as a function of $N$. States separate into two sub-bands due to the dual cavity unit cell.

second unit cell of the crystal and reveals the photonic band structure. Two distinct sub-bands can be identified with noticeably distinct spectral profiles, seen most clearly at $N = 6$ in Fig. 2a. The upper sub-band is located at the higher energy (shorter wavelength) side of the single-cavity state, while the lower sub-band is at the lower energy side. The upper sub-band states are generally well-defined and have a much narrower full-width at half-max (FWHM) compared to the lower energy sub-band, with a value of 10 nm versus 47 nm for the highest energy peaks calculated by multi-peak fitting and highlighted in Fig. 2a. This is due to the direct relationship between the finesse and the mode index, with higher order modes exhibiting lower FWHM[7]. A Peierls photonic band-gap, centered at the single cavity resonance, separates the two sub-bands due to the doubled unit cell. A mid-gap state that should only be expressed in devices with odd numbers of cavities (barring edge effect leakage) is centered on the single-cavity state and is most similar to a crystalline defect due to its symmetry-breaking effect. The origin of these sub-bands is discussed in more detail below and in the SI.

**Density of states**. Varying the mirror thicknesses in the $N = 6$ stack allowed tuning of the density of photonic states

within the band, as illustrated by Fig. 1a. Four devices were constructed by reducing both the silver and aluminum mirror thicknesses equally to 25, 20, 15, and 10 nm. The $N = 6$ data from Fig. 2 was used to complete the set and show the behavior for $d = 30$ nm. The emission spectra show several effects, including a dramatic increase in bandwidth and photonic band gap, a broadening of the peaks, and a noticeable blue-shift in the highest-energy peaks.

Decreasing the thickness of the metallic mirrors results in a broadening of the band as the photonic states approach the extended cavity modes, as seen in Fig. 3a. The energy shift between the perturbed and extended (unperturbed) cavity modes depends on the electric field intensity at the mirror positions; hence, the highest energy mode (top of Fig. 1b) experiences the smallest energy shift due to alignment of the nodes with the metal mirrors. This shift increases as the mode index approaches the fundamental (bottom of Fig. 1b) due to the overlap of antinode positions and the internal mirrors. Decreasing the mirror thickness reduces the perturbation experienced and lowers the energy of the states. The decreased optical pathlength due to removal of mirror material, however, shifts the states to shorter wavelength (higher energy). At higher mode index, the optical pathlength effect dominates due to the lower perturbation experienced by the states, while at lower mode indices this behavior is reversed. The unequal perturbation experienced by the modes results in the observed separation and increased bandwidth in Fig. 3b, c as the mirror thickness drops. Thus, while the highest energy mode experienced a blue shift from 495 to 475 nm, the next lower mode is nearly constant and the four lowest energy modes experienced a red-shift with decreasing mirror thickness, as shown in Fig. 3d.

A second effect of the unequal perturbation experienced by the modes is a net shift of the band center. In the schematic of Fig. 1a, the band center was shown to lie at the energy of the state in the isolated cavities. This resonant energy decreases with the mirror thickness due to increased penetration of the electric field (see Supplementary Fig. 4). In this respect, the observed shift in the photonic band center of the crystal mimics that of the underlying unit cells. The change in the photonic band gap, however, is more closely tied to the overall crystal structure. The band gap is the result of the difference in optical constants between the silver and aluminum mirrors. At zero thickness, these mirrors are indistinguishable and the structure reflects that of the extended cavity in Fig. 1a. Hence, while the band gap appears to increase in Fig. 3d, the ratio of band gap to state separation decreases and approaches unity at $d = 0$ nm.

**Peierls distortion**. The existence of two photonic sub-bands, despite the dimensional equivalence of each cavity within the structure, is a direct consequence of the dimer unit cell configuration employed. The formation of a band gap in a 1D periodic system can be caused by the Peierls distortion, wherein a symmetry-breaking operation results in a doubling of the unit cell[24]. Here, we have deliberately introduced a double unit cell through the alternation of silver and aluminum mirrors for charge injection efficiency considerations. The difference in optical properties between the two mirror types has a dramatic effect on the observed photonic band structure, including the photonic band-gap and the spectral profile of the states.

Much of the difference in optical behavior between the silver and aluminum mirrors can be captured by the penetration depth, the product of the wavelength and the phase shift ($\phi$) given by Eq. (2)[15].

$$\phi = \arctan\left(\frac{n_{org}k_m}{n_{org}^2 - n_m^2 - k_m^2}\right) \qquad (2)$$

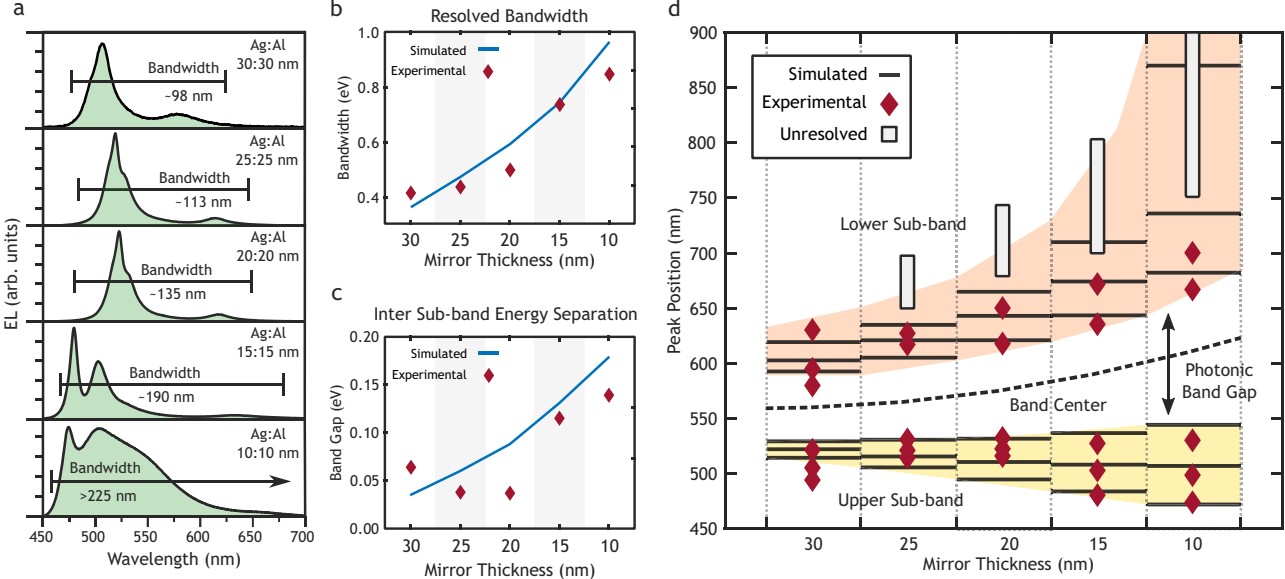

**Fig. 3 Modulation of the density of states through decreasing mirror thickness. a** Experimental emission spectra of six-cavity stacked OLED devices with mirror thickness of $d = 30 \rightarrow 10$ nm. **b** Bandwidth of the resolved peaks as measured between the highest energy state to the first harmonic (second lowest). The fundamental mode of the crystal was not experimentally resolvable for $d < 30$ nm and was omitted for the bandwidth calculation. **c** Average energy separation between states in the upper sub-band. Energy separation increases with decreasing $d$ due decreased perturbation of the extended cavity states. **d** Comparison of experimentally observed and simulated peak locations for $d = 30 \rightarrow 10$ nm. Decreasing $d$ is shown to result in a lower density of states through an increase in the photonic band-gap and the separation between states.

where $n_{org}$ is the index of refraction of the adjacent organic layer and $n_m$ and $k_m$ are the real and imaginary parts of the index of refraction of the metal. Using this equation, the penetration depth of silver at 550 nm (40 nm) is substantially larger than that of aluminum (27 nm) and hence more silver is required to achieve an equivalent absorption. This is offset by the lower real index of refraction for silver (0.06) compared to aluminum (0.79), which affects the optical pathlength through the mirrors[25,26]. The combination of these effects result in the silver and aluminum mirrors playing different roles within the MDC structure.

A series of devices was constructed to examine the roles of the silver and aluminum mirrors within the MDC by keeping the thickness of either the silver anodes or the aluminum cathodes constant while varying the other. We demonstrate the modulation of the Peierls band gap by reducing the thickness of the silver mirrors, beginning with the standard 30:30 nm (Ag:Al) device, as seen in Fig. 4. The electroluminescence spectra shown in Fig. 4a and b reveal that the silver mirror primarily controls the photonic band gap. The theoretical and experimentally measured band gaps are plotted in Fig. 4c (left side). This brings a corresponding shift in the band center as the lower energy states experience a greater change upon perturbation. The aluminum mirrors were found to primarily control the separation of states within the sub-bands and hence the overall bandwidth without significantly affecting the band center, as seen in Fig. 4d and Supplementary Fig. 6.

The difference in behavior between the mirror types can be traced to the unit cell depicted in Fig. 1d. The lower penetration depth and higher losses in the aluminum mirror more strongly pins the electric field and naturally defines the unit cell. This guided the choice of aluminum as the top and bottom mirrors in the even-numbered MDC stacks. In the absence of a silver mirror there exists a single cavity between each set of aluminum mirrors with twice the optical pathlength (see Supplementary Fig. 5). The resonant states of this extended cavity unit cell follow the behavior expressed in Eq. (1) and form a single continuous band

with both the fundamental and the first harmonic mode lying in the visible and near-infrared. Forming a crystal of length $n$ from these extended unit cells would result in each of the states splitting into a band of $n$ hybridized states, as in Fig. 1a.

The addition of a silver mirror perturbs the states within the extended unit cell and brings the fundamental and first harmonic closer in energy. This affects all of the hybridized states within the crystal. The bands formed from these states become more closely spaced and are separated by an energy gap that depends on the degree of perturbation due to the silver mirror. Altering the aluminum mirror thickness changes the underlying states and primarily affects the overall bandwidth and state separation, as shown experimentally in Fig. 4c, d.

## Discussion

We have demonstrated the emergence of photonic band structure in MDCs composed of stacked OLED microcavities and explored methods to control the number of states, density of states, band position, and photonic band gap. Each microcavity added to the crystal was shown to contribute one energy state to the photonic band. As the number of cavities was increased, the states formed a band of discrete photonic states. The bandwidth and density of states was shown to depend on the thickness of the metallic mirrors, while the mismatch in optical properties of the silver and aluminum mirrors was shown to result in a doubling of the unit cell length. Independently varying the silver and aluminum mirrors was shown to produce different effects, with the silver primarily controlling the Peierls bandgap and the aluminum controlling the separation of states within the sub-bands, as the aluminum mirrors define the unit cell.

These results lay the groundwork for a class of electrically-driven photonic crystals and present an avenue for achieving a high degree of control over the emission characteristics of OLED technologies. This may enable a wide range of technologies, such as tunable, low-fidelity frequency combs, narrow bandwidth emitters, wide bandwidth emitters, sensors, and OLEDs with

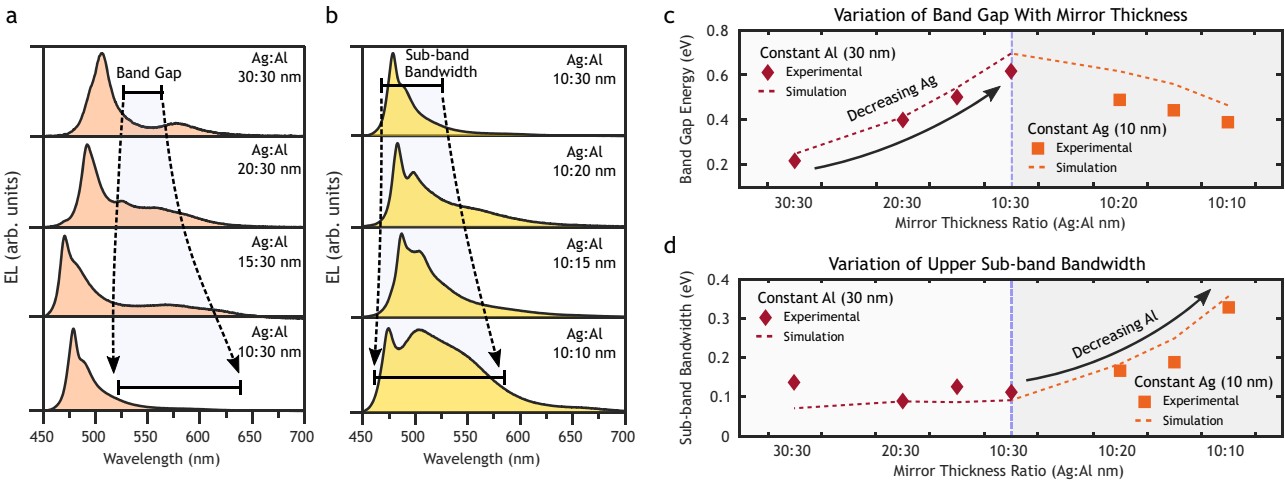

**Fig. 4 Modulation of the photonic band gap by varying one set of mirror thicknesses. a** Emission spectra from constant-thickness Al devices. The silver mirror thickness was reduced from 30 → 10 nm, keeping the Al mirrors fixed at 30 nm. **b** Emission spectra from constant-thickness Ag devices. The aluminum thickness was reduced from 30 → 10 nm, keeping the Ag mirrors fixed at 10 nm. The Ag thickness of 10 nm was selected to enable resolution of the peaks. **c** Modulation of the photonic band gap as a function of the silver and aluminum mirror thicknesses. The silver mirror thickness was found to strongly control the band gap energy. **d** Modulation of the sub-band state separation as a function of the silver and aluminum mirror thicknesses. While the silver mirror thickness was found to have little impact on the separation between states, decreasing the aluminum mirror thickness resulted in an increase in the state separation within the upper sub-band.

specific emission patterns—including the entire white spectrum range.

## Methods

**Device fabrication and characterization**. Each OLED microcavity was fabricated using a three-layer OLED design consisting of 15:1 Ag:Al alloy/MoO$_x$ as the anode, Al/LiF as the cathode, 2,9-Bis(naphthalen-2-yl)-4,7-diphenyl-1,10-phenanthroline2,9-Bis(naphthalen-2-yl)-4,7-diphenyl-1,10-phenanthroline (NBPhen) as the electron transport layer, N,N'-Bis(naphthalen-1-yl)-N,N'-bis(phenyl)-benzidine (NPB) as the hole transport layer and Tris-(8-hydroxyquinoline)aluminum (Alq$_3$) as the emitter layer. The total organics thickness of 115 nm included 70 nm of NPB, 10 nm of Alq$_3$ and 35 nm of NBPhen. A base mirror of 100 nm Al was used in all devices and the LiF and MoO$_x$ injection layers were each 1 nm. Devices were deposited by thermal evaporation onto an 8 nm chromium adhesion layer over thermally oxidized silicon wafers. Emission spectra were recorded using an Ocean Optics Ocean HDX USB spectrometer, with driving current supplied by a Kiethley 2461 sourcemeter. Devices were stored and tested inside a nitrogen glovebox.

## Data availability

The data generated in this study are provided in the Supplementary Information/Source Data file. Source data are provided with this paper.

## Code availability

Code used for modeling is available as a supplementary file, or at devphys.w3.uvm.edu/index/downloads.

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

## Acknowledgements

This material is based upon work supported by the National Science Foundation under Grant No. ECCS-1932677 and No. DMR-1828371. Any opinions, findings, and conclusions or recommendations expressed are those of the authors and do not necessarily reflect the views of the National Science Foundation. Y. Tsuda is grateful for support from the Japan Society for Promotion of Science Overseas Challenge Program for Young Researchers.

## Author contributions

D.A. and M.S.W. wrote and edited the manuscript text and prepared the figures. D.A. developed the computational modeling tool and performed all optical simulations. M.S.W., D.A., B.I., and E.D. developed the experimental procedures. D.A., B.I., E.D., and Y.T. conducted experiments. D.A., B.I., T.Y., and M.S.W. analysed the results. M.S.W. conceived and supervised the project. All authors reviewed the manuscript.

## Competing interests

M.S.W. and D.A. are inventors on PCT Patent Application No. PCT/US21/71679 with the University of Vermont and State Agricultural College as applicant. This patent covers light emitting devices with coupled resonant photonic unit cells.
