## [Peer Review File · Nature Communications]

Emergence and control of photonic band structure in stacked OLED microcavitiesREVIEWER COMMENTS

Reviewer #1 (Remarks to the Author):

This is a well written and structured manuscript about stacked OLEDs that comprise a series of subOLEDs which, when taken together, can exhibit collective optical phenomena. I recommend publication, following taking account of the following comments:

- 1) Authors should comment on complications arising from SPP modes, especially if endeavouring to shift the phenomena to the blue wavelength range
- 2) when authors say "publicly available upon request", I guess that is not what is meant. If publicly available, it would be available to all without a request.
- 3) Please provide more details on the modeling, currently it is tough to fully grasp this aspect.
- 4) Please make explicit if simulations in Fig. 1(a), 1(b), 2(b), 3b,c,d, 4c,d, take non-idealities into account.

Reviewer #2 (Remarks to the Author):

I recommend accepting the paper with minor revision.

The authors studied systematically attributes of multiple units of stacked microcavity OLEDs, presenting metal-dielectric photonic crystal emitting structures. The structures are composed of up to six vertically stacked microcavity OLEDs electrically pumped in parallel. The OLED stack in adjacent cavities was hence, flipped, and with alternating mirrors of Al and Ag, unit cells of two microcavities of opposite orientation were generated. The authors discuss the resulting photonic band structure and the control and effects of added cavities, as well as the thickness and properties of the Al and Ag electrodes on e.g., the photonic bandgap, and number & density of states.

The paper, applying the photonic band gap concept to OLED architectures, is very interesting and clearly written, with the systematic experimental results largely in agreement with simulations.

The authors should address the following:

1. It will be useful to provide some comparative information regarding the luminance or EL of the various structures.
2. What is the effect of the stacked OLED microcavity structure on the angle dependent emission seen in single-unit microcavity OLEDs?
3. It is not clear where the 10 nm and 47 nm FWHM are seen in Fig. 2a for N=6.
4. Check and correct the description in the Methods section where it is stated: "...Ag:Al alloy/MoOx as the cathode, Al/LiF as the anode,..."

Point-by-point response to the reviewer comments

Overview: The reviewers each made 4 suggestions as minor revisions. We have addressed each of these comments and the details are outlined below. Our additional work has resulted in small changes to the main text discussion, two new sections in the supplementary information, and three new figures in the supplementary information comprising 15 total new figure panels.

Comments & Responses

Reviewer 1

1. Authors should comment on complications arising from SPP modes, especially if endeavouring to shift the phenomena to the blue wavelength range

We added a comment on the SPP modes on page 4 (lines 70-73) of the main text.

2. When authors say "publicly available upon request", I guess that is not what is meant. If publicly available, it would be available to all without a request.

We have provided a Source Data file and a Modeling Code file as supplementary information and added a Data Availability and Code Availability statement to the end of the main text (page 7).

3. Please provide more details on the modeling, currently it is tough to fully grasp this aspect.

We have added statements referring the reader to the supplementary information of our most recent publication in Scientific reports. A statement on page 4 (lines 82-83) of the main text refers the reader to the SI of this manuscript, and of the Dahal et al. *Scientific Reports*. We added a similar reference in the SI, on page 1 (line 17). The discussion accompanying the *Scientific Reports* article is quite comprehensive, occupying 4.5 pages of text. But we did not feel that it was necessary or appropriate to reproduce that content in this submission.

4. Please make explicit if simulations in Fig. 1(a), 1(b), 2(b), 3b,c,d, 4c,d, take non-idealities into account.

We added a clarifying statement on page 4 (lines 83-84) of the main text.

Reviewer 2

1. It will be useful to provide some comparative information regarding the luminance or EL of the various structures.

We added a section titled "Luminance Comparison" and one new figure S3 to the Supplementary Information that discusses the comparative EL of the device structures.

2. What is the effect of the stacked OLED microcavity structure on the angle dependent emission seen in single-unit microcavity OLEDs?

We added a section titled "Angle-Resolved Electroluminescence" and two new figures S1 and S2 to the Supplementary Information discussing this topic. We do believe that there is more interesting work to be done on characterizing the behavior of the various photonic states at higher angle, but we are first working to eliminate asymmetries and optical losses in our device stack to obtain better resolution (narrower linewidth) in the lower energy states.

3. It is not clear where the 10 nm and 47 nm FWHM are seen in Fig. 2a for N=6.

We modified Figure 2a to highlight these peaks, modified the figure caption to explain. We added a statement on page 4 (line 107) of the main text to specify that these peaks were extracted by multi-peak fitting.

4. Check and correct the description in the Methods section where it is stated: "...Ag:Al alloy/MoOx as the cathode, Al/LiF as the anode,.."

Checked and corrected, thank you! Main text, page 7 (line 189).

REVIEWERS' COMMENTS

Reviewer #1 (Remarks to the Author):

Manuscript is acceptable

Reviewer #2 (Remarks to the Author):

The authors responded well to all the comments; the paper should now be publish.